# Linking Elastin in Skeletal Muscle Extracellular Matrix to Metabolic and Aerobic Function in Type 2 Diabetes: A Secondary Analysis of a Lower Leg Training Intervention

**DOI:** 10.3390/metabo15100655

**Published:** 2025-10-02

**Authors:** Nicholas A. Hulett, Leslie A. Knaub, Irene E. Schauer, Judith G. Regensteiner, Rebecca L. Scalzo, Jane E. B. Reusch

**Affiliations:** 1Division of Endocrinology, Department of Medicine, University of Colorado School of Medicine (UCSOM), Aurora, CO 80045, USA; leslie.knaub@cuanschutz.edu (L.A.K.); irene.schauer@cuanschutz.edu (I.E.S.); rebecca.scalzo@cuanschutz.edu (R.L.S.); jane.reusch@cuanschutz.edu (J.E.B.R.); 2Veterans Administration Medical Center (VAMC), Aurora, CO 80045, USA; judy.regensteiner@cuanschutz.edu; 3Ludeman Family Center for Women’s Health Research, Department of Medicine, University of Colorado School of Medicine (UCSOM), Aurora, CO 80045, USA; 4General Internal Medicine Division, Department of Medicine, University of Colorado School of Medicine (UCSOM), Aurora, CO 80045, USA

**Keywords:** elastin, collagen, extracellular matrix, insulin sensitivity, skeletal muscle

## Abstract

**Background:** Type 2 diabetes (T2D) is associated with reduced cardiorespiratory fitness (CRF), a critical predictor of cardiovascular disease and all-cause mortality. CRF relies upon the coordinated action of multiple systems including the skeletal muscle where the mitochondria metabolize oxygen and substrates to sustain ATP production. Yet, previous studies have shown that impairments in muscle bioenergetics in T2D are not solely due to mitochondrial deficits. This finding indicates that factors outside the mitochondria, particularly within the local tissue microenvironment, may contribute to reduced CRF. One such factor is the extracellular matrix (ECM), which plays structural and regulatory roles in metabolic processes. Despite its potential regulatory role, the contribution of ECM remodeling to metabolic impairment in T2D remains poorly understood. We hypothesize that pathological remodeling of the skeletal muscle ECM in overweight individuals with and without T2D impairs bioenergetics and insulin sensitivity, and that exercise may help to ameliorate these effects. **Methods:** Participants with T2D (*n* = 21) and overweight controls (*n* = 24) completed a 10-day single-leg exercise training (SLET) intervention. Muscle samples obtained before and after the intervention were analyzed for ECM components, including collagen, elastin, hyaluronic acid, dystrophin, and proteoglycans, using second harmonic generation imaging and immunohistochemistry. **Results:** Positive correlations were observed with elastin content and both glucose infusion rate (*p* = 0.0010) and CRF (0.0363). The collagen area was elevated in participants with T2D at baseline (*p* = 0.0443) and showed a trend toward reduction following a 10-day SLET (*p* = 0.0867). Collagen mass remained unchanged, suggesting differences in density. Dystrophin levels were increased with SLET (*p* = 0.0256). **Conclusions:** These findings identify that structural proteins contribute to aerobic capacity and identify elastin as an ECM component linked to insulin sensitivity and CRF.

## 1. Introduction

Type 2 diabetes (T2D) is a pervasive metabolic disease affecting over 500 million people globally, with prevalence projected to rise over 800 million by 2050 [1]. A hallmark of T2D is the chronic inability to maintain glucose homeostasis, leading to sustained hyperglycemia. Glycemic dysregulation contributes significantly to the development of cardiovascular disease (CVD), which remains the leading cause of mortality in individuals with T2D [2].

One consequence of T2D is a reduction in cardiorespiratory fitness (CRF), which is a strong predictor of cardiovascular and all-cause mortality [3]. In adults and youth with T2D, CRF is reduced compared to body mass index (BMI)-matched controls, suggesting impairments specific to T2D [4]. CRF, measured as maximal oxygen consumption (VO_2peak_), relies upon the integrated capacity of the cardiovascular, pulmonary, and skeletal muscle systems to deliver and utilize oxygen for sustained physical activity. VO_2peak_ is therefore dependent on both ample oxygen delivery via blood flow and sufficient uptake and utilization of delivered oxygen and substrate by the myocytes. Some reports indicate a modest reduction in cardiac output in T2D; however, the cardiac impairment fails to fully explain the defect in CRF [5,6,7]. Therefore, this investigation examines peripheral mechanisms governing oxygen and substrate utilization within the skeletal muscle. In an earlier report, in vivo muscle mitochondrial metabolism, measured using ^31^P magnetic resonance spectroscopy, termed oxidative flux, was revealed to be significantly lower in individuals with T2D compared to overweight controls [8]. However, the provision of supplemental oxygen also corrected the reduced in vivo muscle oxidative flux in individuals with T2D [8]. Also, when mitochondrial respiration was measured ex vivo, under conditions of excess oxygen, differences in those with T2D were no longer apparent [9]. Collectively, these findings suggest that the primary limitation to in vivo skeletal muscle mitochondrial function in T2D is not an intrinsic defect in the mitochondria themselves, but rather intrinsic to the in vivo context—potentially driven by structural differences in the microenvironment surrounding the skeletal muscle.

One potential modifier of maximal oxygen consumption in skeletal muscle is the extracellular matrix (ECM), a structurally and functionally dynamic network composed of collagens, elastin, glycoproteins, proteoglycans, and other matrix-associated proteins. The ECM provides a means of force transduction and mechanical support to muscle fibers and plays a critical role in regulating capillary architecture, interstitial fluid dynamics, insulin signaling, and the diffusion of oxygen and metabolic substrates [10]. Additionally, components of the ECM, such as elastin, can undergo enzymatic degradation to release bioactive fragments—known as elastin-derived peptides—which act as signaling molecules and influence a range of physiological processes including inflammation, angiogenesis, and cellular metabolism [11]. In healthy muscle, the ECM is tightly regulated to allow for efficient transfer of fuel and removal of waste products between the vasculature and myocytes. However, in the context of T2D, the ECM undergoes pathological remodeling characterized by excessive collagen deposition, altered integrin profiles, and increased cross-linking [12,13,14,15,16,17]. These changes can increase tissue stiffness, reduce capillary density, alter signal transduction, and blood flow distribution. Moreover, changes in ECM composition and amount may disrupt the alignment and proximity of capillaries to mitochondria, further compromising oxygen delivery and utilization. Such structural alterations may contribute to the observed deficits in skeletal muscle oxidative metabolism and thereby lower CRF in individuals with T2D. Understanding the extent and functional consequences of ECM remodeling in human skeletal muscle is therefore essential for identifying novel therapeutic targets aimed at restoring muscle and systemic metabolic health.

Accordingly, we hypothesize that pathological remodeling of the skeletal muscle ECM in individuals with T2D is associated with impaired metabolic capacity by limiting oxygen and substrate delivery and utilization, structurally or functionally. We further propose that specific ECM components will correlate with in vivo measures of oxidative extraction and substrate flux, providing insight into the structural and functional constraints on muscle metabolism. Finally, we anticipate that aerobic exercise training will differentially impact ECM composition where elastin and dystrophin will increase while collagen, hyaluronic acid, and proteoglycans will decrease.

## 2. Materials and Methods

### 2.1. Ethical Approval

The University of Colorado Anschutz Medical Campus Institutional Review Board approved this study (IRB# 06–0062) and written informed consent was obtained from all participants. The study conformed to the standards set by the Declaration of Helsinki and was registered on Clinicaltrials.gov (identifier: NCT01793909).

### 2.2. Participants

The present study represents a secondary analysis of previously collected data [18]. This analysis aims to extend the scope of the original investigation by exploring how structural features of the muscle microenvironment may contribute to bioenergetic limitations observed in vivo. Forty-five women and men with (*n* = 21) and without (*n* = 24) T2D between the ages of 30 and 70 years and with a BMI between 25 and 40 kg/m^2^ were enrolled. At baseline, participants were physically inactive, defined as engaging in less than one exercise session per week, confirmed by the Low-level Physical Activity Recall questionnaire [19]. The exclusion criteria included (1) hemoglobin A1c (HbA1c) > 9% (75 mmol/mol), (2) use of insulin, thiazolidinedione, glucagon-like peptide-1(GLP-1) agonists, dipeptidyl peptidase-4 (DPP-4) inhibitors, sodium-glucose cotransporter 2 (SGLT2) inhibitors, or oral steroids, (3) coronary or peripheral arterial disease, cardiac ischemia, conduction abnormalities, beta blocker use, or symptoms limiting exercise, (4) uncontrolled hypertension (systolic blood pressure (BP) > 150 mmHg or diastolic blood pressure > 110 mmHg), (5) obstructive pulmonary disease or asthma, (6) peripheral neuropathy, (7) smoking within the last 2 years, (8) anemia (Hb < 10 mg/dL), (9) autonomic dysfunction (e.g., BP drop > 20 mmHg on standing without HR change), and (10) implanted metal (MRI contraindication). Overweight/obese controls (OWC) without T2D had no health conditions other than excess weight and no more than one first-degree relative with diabetes.

Participants underwent a screening visit to confirm eligibility, followed by four additional study visits before starting SLET. The first visit included consent and a fasting blood sample for a lipid panel, glucose, insulin, and HbA1c. A graded exercise test on a cycle ergometer was performed during the second visit to measure VO_2peak_. During the third visit, a biopsy of the dominant gastrocnemius muscle was performed. The fourth visit included out-of-magnet leg exercise testing combined with near-infrared spectroscopy (NIRS) and plethysmography to measure leg blood flow. The fifth visit involved MRI imaging of the leg for maximal cross-sectional area and in-MRI single-leg exercise testing with ^31^P magnetic resonance spectroscopy (MRS) to assess in vivo mitochondrial function. Visits two to five were spaced 1 to 3 days apart. After SLET, the skeletal muscle biopsy, NIRS data collection, and MRI imaging were repeated with the same intervals between visits.

### 2.3. Single-Leg Exercise Training (SLET)

Samples used to assess the effects of exercise on ECM relationships were available from participants who completed 10 sessions of SLET with their dominant leg (5 days per week for 2 weeks) as described previously [18]. Briefly, each 30–45 min session included six body-weight and resistance exercises: single-legged leg press (3 × 30 reps at 40–50% maximal voluntary contraction), weighted heel raise (3 × 15–20 reps), body-weight heel raises timed for 1 min (3 sets), body-weight heel raises with 5 s pauses (12 reps), body-weight heel raises timed for 30 s, and seated plate dorsiflexion (2 min). Each exercise set included 1 min of rest between repetitions, 3 min of rest between sets, a 5 min warm-up, and a 10 min cool down. SLET has previously been shown to effectively stimulate skeletal muscle adaptations [20]. The high-repetition, low-resistance protocol was designed to engage oxidative metabolism in the lower leg muscles while avoiding a central cardiovascular response. Progression in exercise intensity was made when participants could comfortably complete all sets and repetitions.

### 2.4. Graded Exercise Test

CRF is defined as VO_2peak_ which was determined via graded exercise to exhaustion using a stationary cycle ergometer (Lode Bike, Groningen, The Netherlands) and a metabolic cart (Medgraphics Ultima CPX, Medical Graphics Corp., St. Paul, MN, USA) as described previously [21]. The work rate increased by 10–20 watts per minute, allowing participants to reach peak aerobic power within 8–12 min. Peak VO_2_ was confirmed by a respiratory exchange ratio greater than 1.1. The highest VO_2_ and heart rate, averaged over 15 s during incremental exercise testing, were defined as peak values.

### 2.5. Insulin Sensitivity

Insulin sensitivity is a key factor in T2D severity and may be modulated with ECM. Insulin sensitivity was determined using the hyperinsulinemic euglycemic clamp technique as previous described [22,23]. Briefly, we used dual intravenous catheterization: one for glucose and insulin infusion, and another in a heated dorsal hand vein for arterialized-venous sampling. A descending dose of insulin was administered over the first 10 min (127–40 mU·m^−2^·min^−1^), followed by a continuous insulin infusion (40 mU·m^−2^·min^−1^) from 10 to 180 min. A 20% dextrose solution was titrated to maintain blood glucose at 5 mmol·L^−1^. Blood samples (~1 mL) were collected every 5 min and analyzed immediately. Insulin sensitivity was calculated from the mean glucose infusion rate (GIR) during the final 30 min and expressed per kg of body mass per unit of circulating insulin per min.

### 2.6. Magnetic Resonance Spectroscopy

To investigate skeletal muscle bioenergetics in vivo, we employed ^31^phosphorus magnetic resonance spectroscopy. This technique allows for non-invasive assessment of in vivo mitochondrial oxidative metabolism [18,24].

Acquisition: Imaging and spectroscopy acquisition procedures have been previously described [9,24]. In summary, imaging and MRS were performed using a General Electric 3T magnet with HDx MRI software (version 15M4) and a Siemens 3T magnet with a Skyra platform. The scanners were equipped with a multi-nuclear spectroscopy hardware, research software upgrades, and a custom-built ^1^H/^31^P leg coil (Clinical MR Solutions, Brookfield, WI, USA).

MRS Exercise Protocol: Strength testing of the dominant leg was performed using a custom-built MR-compatible plantar flexion device with force measurement capability as previously described [25,26]. The ^31^P MRS exercise procedure included 60 s of rest, 90 s of isometric plantar flexion at 70% of maximal volitional contraction, and 8 min of recovery. The 90 s isometric exercise bout was chosen for its previously demonstrated use in assessing aerobic and anaerobic processes [27]. Force was continuously monitored and recorded, with verbal and visual feedback provided to maintain the target force.

MRS Analysis: Spectroscopy analysis was performed as previously described [18,24,25,28]. Briefly, peak positions and areas of interest (PCr, Pi, β-ATP, α-ATP, γ-ATP, and phosphomonoester) were determined using time domain fitting with jMRUi [29] and AMARES, which is a nonlinear least-square-fitting algorithm with prior knowledge files [30]. ADP concentrations were calculated using a Michaelis–Menten model of the creatine kinase reaction [28]. Exercise spectra were corrected for saturation using fully relaxed spectra. Calculations utilized data from the end of the exercise and immediate recovery period, including rates of oxidative phosphorylation (OxPhos), initial PCr synthesis (VPcr), and apparent maximum rate of oxidative ATP synthesis (Qmax). Time constants (TC) for ADP and PCr were calculated via regression analyses with Sigmaplot (Version 13.0) (Systat Software, Inc., San Jose, CA, USA). Reported values (VPCr, QMAX, OxPhos, and ADP TC) characterized aspects of oxidative phosphorylation, reflecting the muscle’s ability to generate ATP through oxidative phosphorylation.

### 2.7. Near-Infrared Spectroscopy

To assess muscle oxygenation dynamics during exercise, we utilized NIRS. This technique provides real-time, non-invasive measurements of tissue oxygen saturation and hemodynamics.

Acquisition: NIRS was performed as previously described [31]. Tissue total hemoglobin + myoglobin ([tHb]), deoxy[hemoglobin + myoglobin] ([HHb]), and oxy[hemoglobin + myoglobin] ([OHb]) were assessed using a frequency domain multi-distance NIRS monitor (Optiplex TS, ISS, Champaign, IL, USA) during each constant work rate exercise test. The monitor emitted two wavelengths (690 and 830 nm) and measured absorbance at distances of 2.0, 2.5, 3.0, and 3.5 cm. NIRS data were continuously sampled at 50 Hz and down-sampled to 1 Hz using a running average. During cycling exercise tests, the NIRS probe was positioned on the gastrocnemius of the dominant limb, secured with a Velcro strap, and covered with a cloth bandage to exclude ambient light. The monitor was calibrated before each visit using a calibration phantom with known scattering and optical properties.

NIRS Analysis: Resting values of tissue [tHb], [HHb], and [OHb] were averaged over 30 s before the onset of the exercise. Exercise values were averaged between 270 and 300 s after the onset of the exercise. The absolute change in [tHb], [HHb], and [OHb] from rest- to steady-state exercise was recorded. The change in [tHb] reflects local recruitment of microvascular blood volume/hematocrit. Changes in [HHb] and [OHb] represent local skeletal muscle deoxygenation and oxygen availability, respectively. Data with negative values for any hemoglobin species were discarded.

### 2.8. Second Harmonic Generation Imaging

To assess structural features of the skeletal muscle extracellular matrix, we utilized second harmonic generation (SHG) imaging. This label-free technique enables high-resolution visualization of fibrillar collagen, which is a key ECM component. Gastrocnemius muscle samples were cryopreserved at −80 °C in an optimal cutting-temperature compound and sectioned using a CM3050 S (Leica Biosystems, Deer Park, IL, USA) cryostat at a thickness of 8 µm. ProLong Gold antifade reagent (Thermo P36930, Waltham, MA USA) was applied to each section, followed by the placement of a coverslip prior to imaging.

The SHG imaging system utilized an Olympus FV1000 MPE/DIVER upright microscope (Center Valley, PA USA ) coupled with a Spectra-Physics Insight X3 laser (Milpitas, CA USA ). Imaging was conducted at a laser wavelength of 810 nm with a pulse duration of 48.00 ns. For low-magnification imaging, a 10×/0.50 oil-immersion objective was employed, while a 40×/0.80 oil-immersion objective was used for high-magnification imaging. The microscope was configured to simultaneously capture both forward and backward SHG signals. Two sections of each sample were measured and averaged together for later analysis.

Quantitative image analysis was performed using the FIJI software (version 2.16.0/1.54p). Thresholding techniques were applied to determine collagen area and cross-sectional area, and the square root of the SHG signal was calculated to estimate collagen mass in low-magnification images [32]. High-magnification images were analyzed using CurveAlign to evaluate individual collagen fibers [33].

### 2.9. Immunohistochemistry

To evaluate specific protein components of the skeletal muscle extracellular matrix, we performed immunohistochemistry. This technique enables targeted visualization and quantification of ECM-related proteins within tissue sections. Gastrocnemius muscle samples were sectioned using a CM3050 S cryostat (Leica Biosystems, Center Valley, PA, USA) at a thickness of 8 µm and mounted onto staining slides. The slides were fixed in 4% of formaldehyde and blocked with 1% of bovine serum albumin. Expressions of hyaluronan, elastin, proteoglycans (assessed by wheat germ agglutinin binding to N-acetyl-D-glucosamine), and dystrophin were assessed using biotinylated hyaluronan-binding protein (2.5 µg/mL) (Millipore 385910100UG, Burlington MA USA), elastin (1:100) (Novus Biologicals NB420990, CO USA), wheat germ agglutinin (1:500) (Invitrogen W21405, CA USA), and dystrophin (1:400) (Thermo Fischer FEMA550848, Waltham, MA USA), respectively. Images were captured using a Keyence BZ-X800 microscope (Keyence, Itasca, IL, USA) with a 20× objective and quantified using the FIJI software (version 2.16.0/1.54p). Thresholding techniques were applied and measurements were normalized to the cross-sectional area. Fiber size distribution was assessed with MuscleJ2 v1.0 (ImageJ plugin), which applies automated segmentation algorithms. Images were acquired at identical magnification and resolution across groups. Following preprocessing, elastin staining was used to define fiber boundaries, and MuscleJ calculated the cross-sectional area of each individual fiber [34]. Two sections per sample were averaged for all measurements.

### 2.10. Statistical Analysis

Differences in participant characteristics and baseline collagen metrics were determined with unpaired *t* tests. Data were analyzed with two-way analysis of variance (ANOVA); glycemia (control versus T2D) and training status (pre versus post) were the two factors. Linear regression was determined between ECM and measures of interest. To explore continuous relationships across a broader physiological spectrum, control and T2D groups were pooled for correlation analyses, as they were matched according to BMI and likely share overlapping metabolic characteristics. The level of statistical significance was set at *p* < 0.05 for all data. Data are expressed as mean ± SD. All calculations were performed using GraphPad Prism (version 10.4.1 for Windows, GraphPad Software, San Diego, CA, USA). The datasets generated and/or analyzed during the current study are available from the corresponding author upon reasonable request.

## 3. Results

### 3.1. Participant Characteristics

Baseline characteristics and metabolic profiles are shown in Table 1 [17,29]. No significant differences were observed between the groups in terms of age, systolic blood pressure, diastolic blood pressure, BMI, or body fat percentage. As anticipated, participants with T2D exhibited worse glucose control evidenced by lower glucose infusion rates (GIR) and higher HbA1c compared to the overweight/obesity control (OWC) group (*p* = 0.0008 and *p* < 0.0001, respectively). Blood lipid levels showed a trend towards being higher in the OWC group compared to the T2D group (total cholesterol: *p* = 0.068, LDL: *p* = 0.0874, HDL: *p* = 0.058), potentially attributable to the greater use of statins among the T2D participants (42.9% vs. 8.3%, respectively; *p* < 0.0001). VO_2peak_ was lower in the T2D group compared to the OWC group (*p* = 0.014). VPCr, which quantifies the rate of phosphocreatine recovery in the skeletal muscle following exercise and reflects the mitochondrial oxidative capacity, was lower in those with T2D (*p* = 0.009). Oxyhemoglobin depletion following the beginning of the exercise was not different between groups (*p* = 0.239). Compliance rates for completing single-leg exercise training (SLET) were 93% ± 12% in the OWC group and 87% ± 13% in the T2D group, with no difference in compliance between the groups (*p* = 0.23). There was no significant difference in fiber size between control and T2D groups (*p* = 0.303).

### 3.2. Baseline Collagen Abundance and Organization

Quantification of collagen area (Figure 1A) revealed a greater SHG signal when normalized to cross-sectional area (CSA) in T2D samples compared to controls (*p* < 0.05). However, no significant difference was observed in collagen mass between the groups (Figure 1B) in low-magnification images (Figure 1C). High-magnification images (Figure 1F) were further analyzed using CurveAlign (Figure 1I) to assess collagen organization. Box density, a measure of collagen fiber packing, was significantly lower in T2D samples relative to controls (Figure 1D, *p* < 0.05). In contrast, angle variance (Figure 1E), overall orientation (Figure 1G), and overall alignment (Figure 1H) did not differ significantly between the groups.

### 3.3. Skeletal Muscle Extracellular Matrix Components Before and After Exercise Training

Skeletal muscle extracellular matrix component abundance data are presented in Figure 2. Again, skeletal muscle collagen area relative to cross-sectional area showed a significant effect of diabetes status (*p* = 0.0443) and a trend for training status (*p* = 0.0867) (Figure 2A). Collagen mass, hyaluronic acid, elastin, and proteoglycans relative to cross-sectional area did not exhibit significant differences (Figure 2B–E). Dystrophin relative to cross-sectional area increased with SLET (*p* = 0.0256) (Figure 2F).

### 3.4. Extracellular Matrix Relationships to Clinical Measures

Relationships at baseline between skeletal muscle extracellular matrix components and clinical parameters across both groups are presented in Figure 3 and Appendix A Table A1. There was a relationship between the glucose infusion rate and relative elastin area, with an R^2^ of 0.4259 (*p* = 0.0010) (Figure 3A). The relationship between glucose infusion rate and collagen density had an R^2^ of 0.1189 (*p* = 0.0989) (Figure 3B). Hyaluronan area was related to GIR with an R^2^ of 0.0816 (*p* = 0.1975) (Figure 3C) and related to maximal voluntary contraction with an R^2^ of 0.0901 (*p* = 0.0504) (Figure 3D).

### 3.5. Extracellular Matrix Relationships to Oxidative Dynamics

Relationships between skeletal muscle extracellular matrix components and oxidative dynamic parameters are presented in Figure 4. The relationship between VO_2Peak_ and relative elastin area showed an R^2^ of 0.1923 (*p* = 0.0363) (Figure 4A). The relationship between VO_2Peak_ and relative proteoglycan wheat germ agglutinin area had an R^2^ of 0.1731 (*p* = 0.0541) (Figure 4B). The relationship between VO_2Peak_ and collagen density showed an R^2^ of 0.1189 (*p* = 0.099). The relationship between oxyhemoglobin depletion and relative dystrophin area showed an R^2^ of 0.1791 (*p* = 0.0560) (Figure 4C). The relationship between VPCr and relative dystrophin area was significant, with an R^2^ of 0.4012 (*p* = 0.0036) (Figure 4D). The relationship between oxyhemoglobin depletion and relative dystrophin area showed an R^2^ of 0.1791 (*p* = 0.0560). Lastly, the relationship between VPCr and relative proteoglycan area showed an R^2^ of 0.3351 (*p* = 0.0118) (Figure 4E). A full description of all relationships between each measured component of ECM and clinical parameters can be found in Appendix A Table A1.

## 4. Discussion

In this secondary analysis, we examined skeletal muscle samples from inactive individuals with obesity, with and without T2D, before and after a 10-day SLET. Our objective was to assess changes in the skeletal muscle ECM composition and to explore associations between ECM components and functional measures of oxidative capacity and insulin sensitivity. The analysis revealed that elastin content was positively correlated with GIR, VO_2peak_, and bioenergetics. Dystrophin levels also showed a positive association with VO_2peak_, indicating a link between structural muscle proteins and CRF and bioenergetics. Furthermore, collagen area decreased following SLET, with no difference in mass. Together, these findings suggest that ECM components play a role in skeletal muscle bioenergetics beyond their structural functions.

Our analysis revealed a positive correlation with elastin and insulin sensitivity and a trend toward a negative correlation with collagen density. Collagen and elastin have opposing roles in tissue stiffness [35]. Prior animal studies have shown that hyperglycemia and insulin resistance lead to increased collagen deposition relatively early in disease progression [12,36,37] while elastin may decline [38]. Elastin has been associated with improved tissue repair and a healthier ECM [39,40]. Preclinical studies suggest a dynamic contribution of elastin to insulin action. Elastin degradation products (EDPs) have been associated with insulin resistance in preclinical studies, which is a property that can be reversed using the proteoglycan chondroitin sulfate, supporting a complex interplay between components of the ECM [41]. Insulin and IGF-1 also regulate elastin transcription, further supporting our finding of elastin level correlating with insulin action [42]. In our clinical study, higher elastin levels were linked to greater insulin sensitivity, as indicated by a higher GIR. This may indicate that elastin is a marker of health and losses may drive functional decline. In these participants with obesity and uncomplicated diabetes, there was no difference in elastin by group.

Elastin also may act through binding with other ECM components such as integrins. Emerging evidence suggests that ECM components such as elastin may influence metabolic regulation through integrin-mediated signaling. Integrin-linked kinase, a downstream effector of integrin engagement, has been shown to be essential for maintaining insulin sensitivity in the skeletal muscle, with its disruption leading to impaired insulin-stimulated glucose uptake and systemic insulin resistance [43]. Furthermore, EDPs have been identified as ligands for integrin αVβ3, modulating cellular adhesion and intracellular signaling cascades [44]. Moreover, EDPs may help to explain the observed relationship between elastin and GIR. EDPs have been shown to impair insulin receptor signaling by interacting with the neuraminidase-1 subunit of the elastin receptor complex, leading to desialylation of the insulin receptor β-chain and reduced glucose uptake in the skeletal muscle, liver, and adipose tissue [41,45]. Of note, insulin and IGF-I can stimulate elastin synthesis via PI3K-mediated transcriptional activation, suggesting a feed-forward relationship between ECM remodeling and metabolic regulation [42,46]. These mechanisms highlight the complex interplay between elastin biology and insulin action and support the notion that elastin levels may reflect both structural integrity and metabolic health.

VO_2peak_ correlated positively with elastin and showed a positive trend with proteoglycans by wheat germ agglutinin staining, while no relationship was observed with collagen area or density. Elastin, a critical determinant of passive muscle compliance, imparts elastic recoil properties to skeletal muscle, enabling efficient lengthening and shortening during locomotion. A more compliant ECM may also help preserve capillary integrity and perfusion under mechanical stress, thereby optimizing oxygen delivery to active muscle fibers. These findings underscore the potential importance of ECM composition—particularly elastin—in modulating both the mechanical and metabolic dimensions of muscle function that underpin aerobic performance. Phosphocreatine recovery time, a measure of in vivo mitochondrial respiration, correlates inversely with dystrophin and proteoglycans. These findings support a relationship between the skeletal muscle ECM and bioenergetics. We did not find an association between proteoglycans and cardiorespiratory fitness as reported previously [12]. This may be due to the difference in the muscle sampled. In this study the gastrocnemius is used while previous reports used the quadricep. These muscle groups differ in fiber type and relationship to CRF [42]. A recent study demonstrated that ECM composition in slow- and fast-twitch muscles exhibits differential responsiveness to TGFβ, suggesting that fiber type-specific characteristics may underlie the observed variations [47].

Similarly to prior reports, we found a significantly greater collagen area in participants with T2D compared to overweight controls, whereas in this study we found similar total collagen mass between the groups. This difference between area and mass suggests a reduction in collagen density in T2D, which is a potentially important but understudied aspect of ECM structure. Collagen area showed a downward trend following the 10-day SLET intervention, yet this change did not reach statistical significance when stratified by diabetes status. The relatively short duration of the intervention and small cohort size may have been insufficient to induce measurable changes in collagen abundance. The primary mechanism underlying collagen expansion in the skeletal muscle remains unclear. Prior studies have linked increased collagen deposition to insulin resistance and high-fat feeding in both human and animal models [16,48,49,50]. Moreover, reductions in collagen content following physical exercise and bariatric surgery suggest that adiposity may be a key driver of ECM accumulation [51]. Conversely, other work has implicated hyperglycemia as a contributor to ECM expansion, as ECM excess resolved when hyperglycemia was treated [12]. In our study, participants with T2D had relatively well-controlled glycemia, with an average HbA1c of 6.3%. Despite an average diabetes duration of nearly nine years, the absence of pronounced hyperglycemia may explain the lack of significant differences in collagen remodeling. It is possible that both pronounced and sustained elevations in blood glucose are required to drive more substantial ECM changes. Furthermore, statin use is more prevalent among individuals with T2D in this study, which may have influenced ECM characteristics. Evidence suggests that statins are associated with skeletal muscle pathologies, including disrupted sarcomeric organization, reduced force production, and increased generation of reactive oxygen species [52,53]. These effects may indirectly impact ECM, although the specific influence of statins on ECM remodeling remains unclear. The addition of statins to the T2D muscle microenvironment introduces a potentially confounding factor that warrants further investigation. However, the relationship of elastin to VO_2peak_ and GIR did not change when participants using statins were excluded. Additionally, the use of overweight controls may have masked group differences; the inclusion of a lean comparator group may have revealed more distinct ECM alterations.

The physiological implications of decreased collagen density warrant further investigation, as it is possible that the observed differences may be attributable to shifts in the proportional abundance of collagen types and/or organization. We explored whether this was associated with changes in alignment by higher magnification SHG and did not observe differences in collagen organization. SHG imaging excels at measuring fibrillar collagen types but is not as sensitive at measuring other types of collagen [32]. Alterations in the relative abundance of collagen types can have profound effects on cell–matrix interactions [54], angiogenesis [55], and tissue elasticity [56], potentially influencing both signaling and functional properties of the skeletal muscle in individuals with T2D. Accordingly, further investigation into nonfibrillar collagen subtypes, such as collagen type IV, is warranted to elucidate their potential involvement.

One of the more unexpected findings from this study was the relationship between dystrophin and skeletal muscle bioenergetics. Given dystrophin’s established role in maintaining ECM integrity and preventing actin depolymerization [57], we anticipated the observed increase in response to exercise training. However, contrary to expectations, a lower dystrophin area was associated with higher phosphocreatine turnover (VPCr), suggesting improved mitochondrial efficiency. This contradictory relationship underscores the complexity of ECM remodeling in metabolic adaptation.

It has been established that low levels of dystrophin are associated with insulin resistance, T2D, and impaired vasodilation in humans [58,59,60]. In MDX mice, dystrophin deficiency leads to a change in the nitric oxide synthase subtype from endothelial to inducible. This shift raises intracellular calcium concentrations, disrupting glucose metabolism through reduced glycogen phosphorylase activity and increased GSK3β signaling [61,62]. These calcium-mediated effects may also contribute to mitochondrial dysfunction and oxidative stress [62]. While these findings highlight a potential mechanism for low levels of dystrophin to influence CRF and insulin sensitivity, our data suggest that the relationships between dystrophin levels and bioenergetic processes are not linear, and the relative contribution of each component may not independently predict a clinical endpoint such as VPCr.

Interestingly, we did not observe a significant association between dystrophin levels and glucose infusion rate, suggesting that dystrophin’s role in glycemic control may be context-dependent or influenced more robustly by other regulatory mechanisms. T2D status did not appear to drive dystrophin expression, as no differences were observed between the groups. However, obesity remains a potential factor that warrants further investigation.

## 5. Conclusions

In summary, this study highlights the novel relationships between individual skeletal muscle ECM components and key clinical measures of metabolic health and functional capacity. Among these, elastin emerged as a particularly compelling candidate, showing strong positive associations with both insulin sensitivity and CRF. These findings suggest that elastin may be an underrecognized and a potentially powerful marker of skeletal muscle health. We also observed disease specific alterations in collagen and dystrophin content associated with T2D and exercise training, reinforcing the diverse roles of ECM components in muscle physiology. These findings underscore the importance of moving beyond bulk ECM assessments to investigate specific components in relation to metabolic outcomes. Future research should aim to uncover the molecular mechanisms driving these associations, with a focus on the regulation of elastin and its contribution to skeletal muscle metabolic health and whether these changes actively mediate improvements in insulin sensitivity and fitness will be critical for translating these findings into therapeutic strategies.

## Figures and Tables

**Figure 1 metabolites-15-00655-f001:**
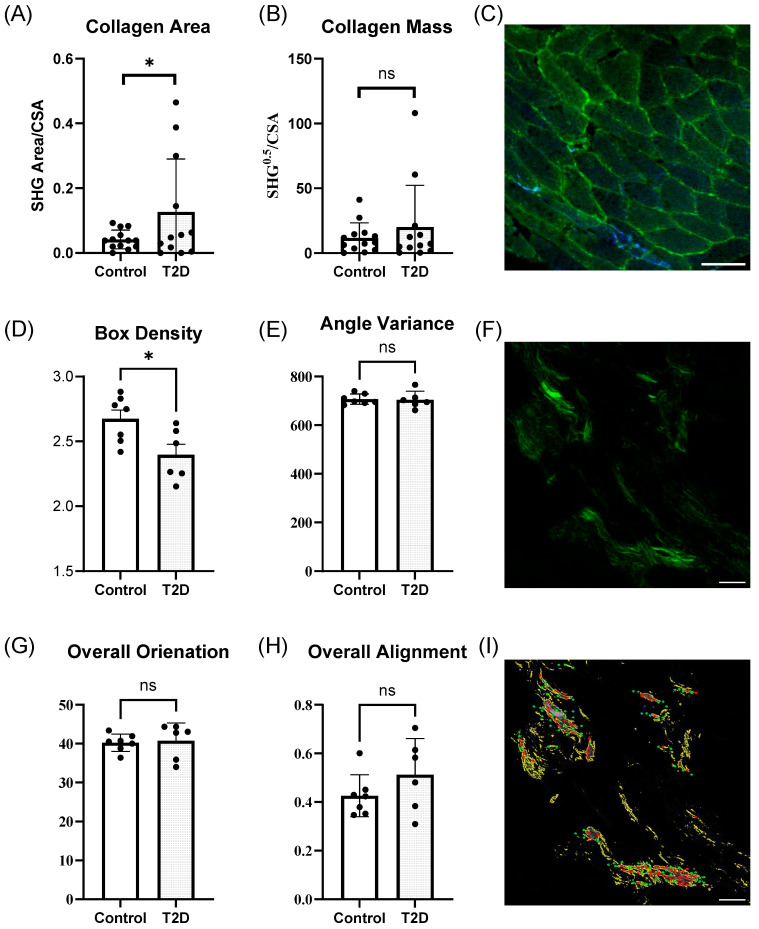
Collagen content at baseline in type 2 diabetes and controls. (**A**), Skeletal muscle box density was significantly higher in control subjects than those with T2D (*p* = 0.0229). (**B**), Collagen mass relative to cross-sectional area (no significant difference). (**C**), Representative image of low-magnification muscle section. Green is the forward scatter representing collagen and blue is the backward scatter; scale bar is 100 µm. (**D**), Box density (*p* = 0.0229). (**E**), Angle variance (no significant difference). (**F**), Representative image of high-magnification muscle section. Green is the forward scatter representing collagen. Scale bar is 10 µm. (**G**), Overall orientation score (no significant differences). (**H**), Overall alignment score (no significant differences). (**I**), Representative image of high magnification after analysis with CurveAlign. Yellow lines indicate collagen strands, red dots show fiber location, and green lines show the orientation. Scale bar is 10 µm. Only baseline samples are used in this analysis. High-magnification images were captured from five distinct regions of each tissue section, and measurements were averaged to obtain a representative value for each sample. * = *p* > 0.05 ns = not significant.

**Figure 2 metabolites-15-00655-f002:**
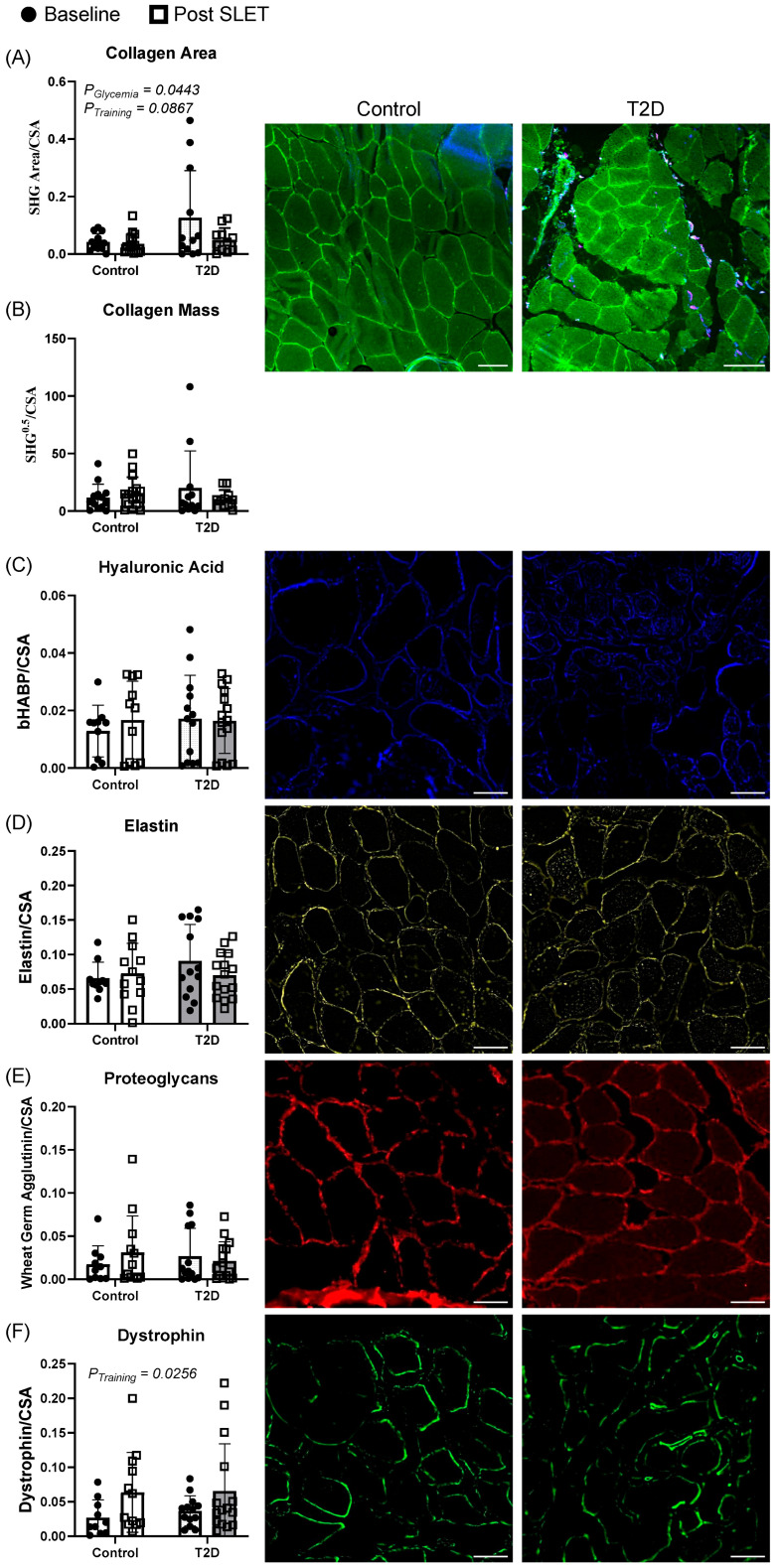
Skeletal muscle extracellular matrix components in type 2 diabetes and exercise training. (**A**), Skeletal muscle collagen area relative to cross-sectional area (*p* = 0.0443 for the effect of glycemia and *p* = 0.0867 for the effect of training status). (**B**), Collagen mass relative to cross-sectional area (no significant differences). (**C**), Hyaluronic acid relative to cross-sectional area (no significant differences). (**D**), Dystrophin relative to cross-sectional area (*p* = 0.0256). (**E**), Elastin relative to cross-sectional area (no significant differences). (**F**), Proteoglycans relative to cross-sectional area (no significant differences). All scale bars are 100 µm. SHG = second harmonic generation. CSA = cross-sectional area.

**Figure 3 metabolites-15-00655-f003:**
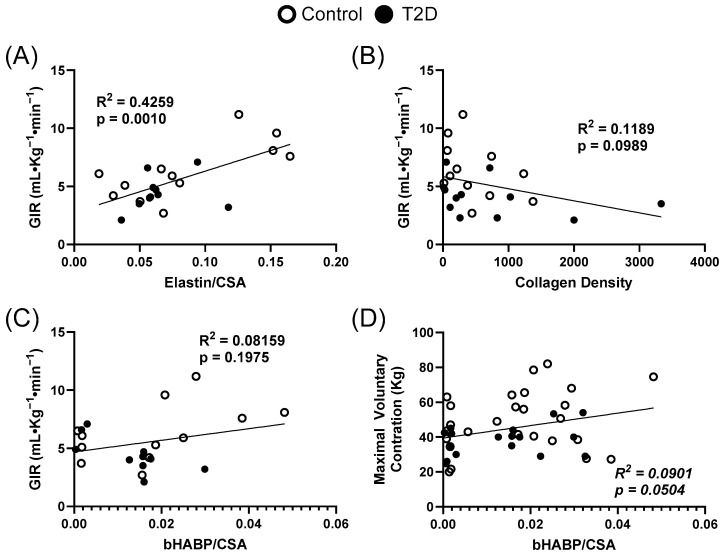
Relationships between skeletal muscle extracellular matrix components and clinical parameters. (**A**), Relationship between glucose infusion rate and relative elastin area (R^2^ = 0.4259 *p* = 0.0010). (**B**), Relationship between glucose infusion rate and collagen density (R^2^ = 0.1189 *p* = 0.0989). (**C**), Relationship between glucose infusion rate and relative hyaluronan area (R^2^ = 0.1975 *p* = 0.1975). (**D**), Relationship between maximal voluntary contraction and relative hyaluronan area (R^2^ = 0.0901 *p* = 0.0504). CSA = cross-sectional area.

**Figure 4 metabolites-15-00655-f004:**
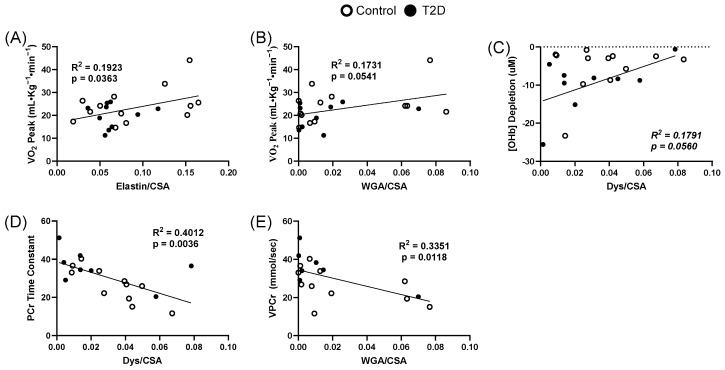
Relationships between skeletal muscle extracellular matrix components and oxidative dynamic parameters. (**A**), Relationship between VO_2Peak_ and relative elastin area (R^2^ = 0.1923 *p* = 0.0363). (**B**), Relationship between VO_2Peak_ and relative proteoglycan area (R^2^ = 0.1731 *p* = 0.0541). (**C**), Relationship between oxygen hemoglobin depletion and relative dystrophin area (R^2^ = 0.1791 *p* = 0.0560). (**D**), Relationship between VPCr and relative dystrophin area (R^2^ = 0.4012 *p* = 0.0036). (**E**), Relationship between VPCr and relative proteoglycan area (R^2^ = 0.3351 *p* = 0.0118). CSA = cross-sectional area. WGA = wheat germ agglutinin. Dys = dystrophin. VPCr = velocity of phosphocreatine turnover.

**Table 1 metabolites-15-00655-t001:** Participant characteristics at baseline: data are presented as means ± SD. * *p* > 0.05. GIR = glucose infusion rate; VPCr = rate of phosphocreatine synthesis.

	Control (*n* = 24)	T2D (*n* = 21)
Age (y)	52 ± 13	61 ± 9
Sex (% Female)	45.8	47.4
BMI (Kg/m^2^)	30.0 ± 4.6	31.9 ± 8.0
Body Fat (%)	34.2 ± 8.2	37.2 ± 8.0
SBP (mmHG) *	118.8 ± 11.7	125.0 ± 11.0
DBP (mmHG)	80.3 ± 8.0	79.5 ± 7.1
Duration of DM (y) *	0 ± 0	8.8 ± 7.0
HbA1c (%) *	5.3 ± 0.3	6.3 ± 0.7
Current Metformin Use (%) *	0	88.9
Total Cholesterol (mmol/L) *	4.544 ± 0.854	4.042 ± 0.885
HDL (mmol/L)	1.311 ± 0.287	1.161 ± 0.331
LDL (mmol/L)	2.848 ± 0.799	2.359 ± 0.786
Current Statin Use (%) *	8.3	42.9
GIR (mL/Kg·min) *	7.1 ± 2.3	4.8 ± 1.9
VO_2Peak_ (mL/Kg·min) *	24.5 ± 7.8	18.7 ± 4.8
VPCr (mmol/s) *	0.223 ± 0.088	0.155 ± 0.077
Oxyhemoglobin Depletion (µM)	−3.4 ± 5.3	−6.2 ± 7.0
Maximal Voluntary Contraction (Kg)	43.2 ± 15.2	39.0 ± 8.0
Fiber Size (µm)	184.1 ± 33.6	206.1 ± 35.4

## Data Availability

The original contributions presented in this study are included in the article. Further inquiries can be directed to the corresponding author.

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
