# Peer review of "Linking Elastin in Skeletal Muscle Extracellular Matrix to Metabolic and Aerobic Function in Type 2 Diabetes: A Secondary Analysis of a Lower Leg Training Intervention"

_metabolites, 2025, doi:10.3390/metabo15100655_

Round 1

Reviewer 1 Report

Comments and Suggestions for Authors

I would like to thank the authors for the opportunity to review this manuscript and to commend them for addressing a highly relevant topic with scientific rigor. The detailed comments and suggestions are provided in the attached document, with the aim of supporting the improvement of the text and further strengthening the presentation of the results.

Sincerely.

Comments on the Quality of English Language

The language is clear and appropriate for scientific communication, with no improvements necessary. However, care should be taken with abbreviations.

Reviewer 2 Report

Comments and Suggestions for Authors The authors assessed changes in skeletal muscle ECM composition and explored associations between ECM and oxidative capacity and insulin sensitivity. The authors should address the following comments to improve the manuscript:   1. The Abstract states T2D n = 12 and control n = 15, but Table 1 and the Results section report control n = 24 and T2D n = 21.   2. In sections 3.4 and 3.5, the authors investigated the relationships between ECM and clinical and oxidative parameters. I am very confused as to why pooling control and T2D together is rational. The authors should explain the assumptions and rationale for pooling the two groups together.

Round 2

Reviewer 1 Report

Comments and Suggestions for Authors

Dear colleagues, I hope you are well.
Please find my contributions below.
Kind regards

Author Response

Comment: 

The manuscript entitled: "Linking elastin in the extracellular matrix of skeletal
muscle to metabolic and aerobic function in type 2 diabetes: a secondary
analysis of a structured exercise intervention
After a detailed reading of each section of the manuscript, advances are noted in relation
to the previous version, with improvements in the clarity of the title, problematization in
the abstract, methodological details, and deeper discussion. The conclusion is also more
concise.
However, the title of the manuscript still needs to be revised, as the term “structured
exercise” sounds redundant—every exercise protocol presupposes some degree of
structure. It is recommended that the authors clarify this point by replacing the expression
with a more objective description of the intervention model used (e.g., “lower limb
training” or “single-leg exercise training”), which would give the title greater scientific
precision. Another point to highlight if the authors intend to modify the title is the
alignment with the manuscript’s objective.

Response: 

Thank you for your thoughtful feedback and for recognizing the improvements made in the revised manuscript. We appreciate your suggestion regarding the manuscript title and agree that the term “structured exercise” may be redundant. To enhance clarity and scientific precision, we have revised the title to better reflect the specific intervention model used in the study.

We revised to the following title:

Linking Elastin in Skeletal Muscle Extracellular Matrix to Metabolic and Aerobic Function in Type 2 Diabetes: A Secondary Analysis of a Lower Leg Training Intervention